# BARS Influences Neuronal Development by Regulation of Post-Golgi Trafficking

**DOI:** 10.3390/cells11081320

**Published:** 2022-04-13

**Authors:** Laura Gastaldi, Josefina Inés Martín, Lucas Javier Sosa, Gonzalo Quassollo, Yael Macarena Peralta Cuasolo, Carmen Valente, Alberto Luini, Daniela Corda, Alfredo Cáceres, Mariano Bisbal

**Affiliations:** 1Instituto de Investigación Médica Mercedes y Martín Ferreyra (INIMEC), Consejo Nacional de Investigaciones Científicas y Técnicas (CONICET), Universidad Nacional de Córdoba, Córdoba 5016, Argentina; gastaldi.laura@gmail.com (L.G.); joinesmartin@gmail.com (J.I.M.); gquassollo@immf.uncor.edu (G.Q.); acaceres@immf.uncor.edu (A.C.); 2Centro Investigación Medicina Traslacional Severo R Amuchástegui (CIMETSA), Instituto Universitario Ciencias Biomédicas Córdoba (IUCBC), Córdoba 5016, Argentina; 3Departamento de Química Biológica Ranwell Caputto, Facultad de Ciencias Químicas, CIQUIBIC-CONICET-Universidad Nacional de Córdoba, Córdoba 5016, Argentina; lucas@fcq.unc.edu.ar (L.J.S.); yael.peralta.cuasolo@mi.unc.edu.ar (Y.M.P.C.); 4Institute of Experimental Endocrinology and Oncology “G. Salvatore”, National Research Council, 80131 Naples, Italy; carmen.valente@cnr.it (C.V.); alberto.luini@cnr.it (A.L.); 5Department of Biomedical Sciences, National Research Council, 00185 Rome, Italy; daniela.corda@cnr.it; 6Instituto Universitario Ciencias Biomédicas de Córdoba (IUCBC), Córdoba 5016, Argentina

**Keywords:** neurons, neuronal development, membrane trafficking, Golgi apparatus, fission, BARS

## Abstract

Neurons are highly polarized cells requiring precise regulation of trafficking and targeting of membrane proteins to generate and maintain different and specialized compartments, such as axons and dendrites. Disruption of the Golgi apparatus (GA) secretory pathway in developing neurons alters axon/dendritic formation. Therefore, detailed knowledge of the mechanisms underlying vesicles exiting from the GA is crucial for understanding neuronal polarity. In this study, we analyzed the role of Brefeldin A-Ribosylated Substrate (CtBP1-S/BARS), a member of the C-terminal-binding protein family, in the regulation of neuronal morphological polarization and the exit of membrane proteins from the Trans Golgi Network. Here, we show that BARS is expressed during neuronal development in vitro and that RNAi suppression of BARS inhibits axonal and dendritic elongation in hippocampal neuronal cultures as well as largely perturbed neuronal migration and multipolar-to-bipolar transition during cortical development in situ. In addition, using plasma membrane (PM) proteins fused to GFP and engineered with reversible aggregation domains, we observed that expression of fission dominant-negative BARS delays the exit of dendritic and axonal membrane protein-containing carriers from the GA. Taken together, these data provide the first set of evidence suggesting a role for BARS in neuronal development by regulating post-Golgi membrane trafficking.

## 1. Introduction

The C-terminal-binding protein (CtBP) family, encoded by two genes designated as CtBP1 and CtBP2, has been implicated in development, differentiation, oncogenesis, and apoptosis acting in the nucleus as transcriptional co-repressors [1]. BARS (Brefeldin A ADP-Ribosylated Substrate), a short form of CtBP1, has also been identified as a key player in membrane fission events including macropinocytosis, fluid-phase endocytosis, COPI-coated vesicle formation, post-Golgi carrier formation, and Golgi ribbon partitioning during mitosis [2]. The mechanism of action of BARS in fission has been studied in cultured cell lines as a mediator of basolateral post-Golgi carrier biogenesis. In polarized epithelial cells (e.g., MDCK cells), expression of BARS dominant-negative (DN) mutants or RNA interference leads to inhibition of vesicular stomatitis virus (VSVG) trafficking to the basolateral (BL) membrane; in these cells, tubular carriers containing VSVG cargo fail to detach from the Trans Golgi Network (TGN), blocking transport destined to the BL surface [3]. BARS-mediated fission of BL tubular carriers at the TGN appears to be dynamin-independent, involving the assembly of a complex that includes ARF, frequenin, the phosphoinositide kinase PI4KIII, 14-3-3γ, and the kinases PKD and PAK [3,4,5].

In situ hybridization studies have demonstrated strong expression of CtBP family members in the nervous system [6]. Previous studies also showed that CtBP1 and CtBP2 display highly specific distribution in the adult brain, differing in their expression levels, regional and cell-specific expression patterns, and subcellular targeting [7]. Analysis of mice carrying single or double deletions of CtBP genes indicates that these proteins have both redundant and non-overlapping functions; interestingly, double knock out of CtBP1 and CtBP2 in mice results in delayed development of the forebrain and midbrain. It has been proposed that this phenotype results from changes in the activities of transcription factors affected by CtBP family member depletion [8]. However, whether BARS regulates neuronal development by modulating membrane trafficking remains to be established. With this consideration in mind and to gain further insights into the neuronal functions of CtBP proteins, in the present study we examined the consequences of the up- or down-regulation of BARS on neuronal development and the trafficking of axonal and dendritic membrane proteins. The results obtained suggest that by regulating post-Golgi membrane trafficking, BARS influences several aspects of nerve cell development, including axon-dendrite extension/branching in cultured neurons, as well as neuronal cortical migration and the multipolar–bipolar transition in cortical neurons that develop in situ.

## 2. Materials and Methods

### 2.1. Animal Use and Care

Pregnant Wistar rats and C57bl/6 mice were born in the vivarium of INIMEC-CONICET-UNC (Córdoba, Argentina). Wistar rat lines were originally provided by Charles River Laboratories International Inc. (Wilmington, NC, USA). C57bl/6 mice lines were originally provided by Laboratorio de Animales de Experimentación (LAE, Facultad de Ciencias Veterinarias, Universidad Nacional de La Plata, Buenos Aires, Argentina). All procedures and experiments involving animals were approved by the Animal Care and Ethics Committee (CICUAL http://www.institutoferreyra.org/en/cicual-2/, accessed on 1 April 2016) of INIMEC-CONICET-UNC (Resolution numbers 014/2017 B, 015/2017 B, 006/2017 A, and 012/2017 A) and were in compliance with approved protocols of the National Institute of Health Guide for the Care and Use of Laboratory Animals (SENASA, Argentina). 

All methods were carried out in accordance with relevant guidelines and regulations.

### 2.2. DNA Constructs

The following cDNA constructs were used in this study: (1)cDNA coding BARS wild type (WT), the point mutated BARS D355A, a fission dominant-negative mutant [3], and BARS G172E, a mutant with a point mutation in the nucleotide-binding site (NBD) that disrupts the dimer formation and the loss of transcriptional co-repression activity [2,9], were generous gifts from Dr. Alberto Luini (Institute of Biochemistry and Cell Biology, National Research Council, Naples, Italy).(2)cDNAs coding the point mutated BARS S147A fission dominant-negative mutant [4], and the BARS double mutants S147A D355A and S147A G172E, were generated using the QuikChange Site-Directed-Mutagenesis Kit (Stratagene) with the following primers: S147AF 5′ *GGCACTCGGGTCCAGGCTGTAGAGCAGATCCG 3*′ and S147AR 5′ *CGGATCTGCTCTACAGCCTGGACCCGAGTGCC 3*′.(3)cDNAs coding for the Golgi-resident galactosyl-transferase 2 (GalT2) fused to the NH2-terminal domain of EYFP or m-Cherry (GalT2-YFP and GalT2-mCherry) were a generous gift from Dr. Jose L. Daniotti and Dr. Hugo Maccioni, CIQUIBIC-CONICET, Argentina; see also [10].(4)FM plasmids: cDNA sequences from type I transmembrane proteins (ApoER2, L1, and p75NTR) were PCR amplified and then cloned into the plasmid pFM4-GFP type I, which has the human growth hormone signal sequence with the following primers: for ApoER2 FM-ApoER2-F 5′ *GACTAAGCTTTACCCGTACGACGTCCCGGAC* 3′ and FM-ApoER2-R 5′ *CTGAGTCGACTGGGGTAGTCCATCATCTTCAAG* 3′; for L1 FM-L1-F 5′ *GACTAAGCTTATCCAGATCCCCGAGGAATATG* 3′ and FM-L1-R 5′ *CTGAACCGGTATTTCTAGGGCCACGGCAGGG3*’; and for p75 FM-p75-F 5′ *GACTGAATTCCAAAGGAGGCATGCCCCACAGG* 3′ and FM-p75-R 5′ *CGTAGGATCCGTGGATGTGGCAGTGGACTCAC* 3′. A cDNA sequence from TfR, a type II transmembrane protein, was PCR amplified and then cloned into the plasmid pFM4-GFP type II with the following primers: FM-TfR-F 5′ *GATCAGATCTGCCACCATGGATCAAGCTAGATCAGC* 3′ and FM-TfR-R 5′ *CGTAGAATTCGAAACTCATTGTCAATGTCCC* 3′. pFM4-GFP type I and pFM4-GFP type II were a generous gift from Dr. Enrique Rodriguez-Boulan (Margaret Dyson Vision Research Institute, Weill Cornell Medical College, USA).(5)A BARS short hairpin (sh) RNA (shRNA-BARS) and its corresponding scrambled control sequence (sc-shRNA-BARS) were constructed using previously described procedures [11]. In brief, DNA fragments containing U6-sh-RNA and U6-sh-scrambled were inserted into pCAG vector in which the GFP (pCAGIG) or HcRed (pCA-HcRed) cDNA is under the control of a chick actin-minimal (CAG) promoter, and 5′ *GGGTCCAGAGTGTAGAGCAGA* 3′ wasa used as the target sequence. BLAST analysis of target sequence showed 100% homology to rat and mouse.(6)For in utero electroporation BARS S147SA G172E were subcloned into pCAGIG plasmid using XhoI/EcoRI restriction sites and in pCAGGS-IRES-RFP plasmid (a generous gift from François Guillemot, The Francis Crick Institute, London, UK) using an EcoRV restriction site. pCA-ApoER2-GFP were cloned from pEGFP-ApoER2 (provided by Dra. María Paz Marzolo, Pontificia Universidad Católica de Chile, Chile) in the pCAGIG vector using the AgeI/BsrGI restriction site. All plasmids were verified by DNA sequencing (Macrogen Inc., Seoul, Korea).

### 2.3. Affinity Purification of Anti-BARS Antibody

An amount of 1 mg of His-BARS was resuspended in 2 mL of PBS. Two milliliters of complete Freund’s adjuvant (Sigma-Aldrich, St. Louis, MO, USA) was added, and this mixture was used to immunize one rabbit (New Zealand strain). The rabbit was boosted after 21 and 42 days with 1 mg of antigen containing the same volume of incomplete Freund’s adjuvant. After collecting the blood, it was allowed to clot at 37 °C for 60 min, and then kept O/N at 4 °C to allow the clot to contract. The serum was removed from the clot and the insoluble material by centrifugation at 10,000× *g* for 10 min at 4 °C, and stored at −80 °C. 

The affinity purified antibody was obtained by loading purified IgGs into a protein-A-Sepharose column with covalently linked GST-BARS antigen. Then, 500 mg of total IgGs suspended in 5 mL of distilled water were packed into a column and washed with 100 mL of distilled water under a constant flow. The beads were washed with 30 mL of PBS, and 2 mL of antiserum was loaded onto the column at 0.5 mL/min using FPLC system (Pharmacia BioTech, Buckinghamshire, UK). After column washing with 30 mL of PBS, the retained IgGs were eluted with 15 mL of 0.1 M glycine, pH 2.5. Fractions of 0.5 mL were collected, and their protein content quantified by spectrophotometric analysis using a protein assay kit (Bio-Rad, Hercules, CA, USA). The six fractions containing the highest concentrations of protein were pooled and neutralized with 1 M Tris, pH 11. The total IgGs were loaded into a column to which the recombinant GST-BARS was covalently coupled. The anti-BARS antibody was purified through this column with a procedure analogous to that given for the total IgGs. Around 50 µg to 70 µg of specific IgGs were obtained from each 1.0 mL serum.

### 2.4. Culture, Transfection, and Immunofluorescence

Embryonic day-18 rat embryos (euthanized by CO_2_ overdose) were used to prepare primary hippocampal cultures as previously described [12,13]. Briefly, hippocampi from 18 day fetal rats were dissected and incubated with trypsin (0.25% for 15 min at 37 °C) (Thermo Fisher Gibco, Waltham, MA, USA; Cat Number: 15090-046) and mechanically dissociated by trituration with a Pasteur pipette. Cells were plated on Cover Glasses, Circles, 12 mm (Marienfeld Superior, Lauda-Königshofen, Germany; Cat Number: 633029), coated with 1 mg/mL poly-l-lysine (Sigma Chemical Co., St. Louis, MO, USA; Cat Number: P2636) at a density of 2000 cells/cm^2^ in minimum essential medium (MEM, Thermo Fisher Gibco; Cat Number: 61100-061) supplemented with CTS GlutaMAX I Supplement (Thermo Fisher Gibco; Cat Number: A1286001), Sodium Piruvate (Thermo Fisher Gibco; Cat Number: 11360070), Penicillin-Streptomycin (Thermo Fisher Gibco; Cat Number: 15140122), and 10% horse serum (Thermo Fisher Gibco; Cat Number: 16050122). After 2 h, the coverslips were transferred to dishes containing serum-free Neurobasal (Thermo Fisher Gibco; Cat Number: 21103049) with B-27 Plus Supplement (Thermo Fisher Gibco; Cat Number: A3582801) and CTS GlutaMAX I Supplement (Thermo Fisher Gibco; Cat Number: A1286001).

Cultured neurons were transfected with Lipofectamine 2000 (Thermo Fisher Gibco; Cat Number: 11668027), following manufacturer’s instructions. Neurons were fixed with 4% *w*/*v* paraformaldehyde (Sigma-Aldrich, Cat Number: 441244) and 4% sucrose diluted in phosphate buffered saline (PBS) for 20 min at room temperature (RT) as described [12]. Fixed cells were washed 3 times with PBS, permeabilized in 0.2% Triton X-100 in PBS at RT for 5 min, and again washed in PBS before antibody incubation. Cells were incubated in blocking buffer (bovine serum albumin 5%/PBS) for 1 h and incubated for 1 h at RT with primary antibodies, washed with PBS, and then incubated with fluorescent-conjugated secondary antibodies. Cells were then washed with PBS and the coverslips mounted using FluorSave (Millipore Calbiochem, Burlington, MA, USA; Cat Number: 34-578-9).

The following primary antibodies were used for immunofluorescence (IF) in this study: a monoclonal antibody (mAb) against GM130, a marker of the *cis*-Golgi compartment (BD Biosciences, San Jose, CA, USA, Cat Number: 610823, RRID:AB_398142) diluted 1:250; an mAb against TGN38, a marker of Golgi compartment (BD Biosciences, Cat Number: 610898, RRID:AB_398215) diluted 1:250; an mAb against tau protein (clone Tau-1; Millipore, Cat Number: MAB3420, RRID: AB_94855) diluted 1:1000; an mAb against MAP2 (clone AP-20; Sigma Aldrich Cat Number: M1406. RRID: AB_477171); an mAb against βIII-tubulin, a neuronal differentiation marker (clone 2G10, Abcam, Cambridge, UK, Cat Number: ab78078, RRID: AB_2256751) diluted 1:500; a polyclonal anti-Ki67, a proliferation marker (Abcam, Cat Number: ab16667, RRID: AB_302459) diluted 1:250; an mAB against γ-tubulin protein (clone GTU-88, Sigma Aldrich, Cat Number: T6557, RRID: AB_477584) diluted 1:500; a polyclonal anti-BARS antibody against BARS protein developed and provided by Dr. Alberto Luini, diluted 1/50 for endogenous BARS IF, 1/3000 for BARS overexpressed IF. Secondary antibodies conjugated with fluorochromes (AlexaFluor 488 nm, 568 nm, and 633 nm; Molecular Probes, Eugene, OR, USA, Cat Numbers: A11029, A11006, A11077, A11004, A21070 and A21052) were diluted 1/1000.

Cells were visualized using spectral LSM 800 (Carl Zeiss, Gina, Germany) or FV1000 (Olympus, Tokyo, Japan) inverted confocal microscopes. Images were processed using Adobe Photoshop. Post-imaging analysis and measurements were conducted using Fiji-Image J 1.53v software (NIH, Bethesda, MD, USA).

### 2.5. Regulated Secretion/Aggregation Protocol

Synchronization assay to study the biosynthetic route consists of the expression of a cDNA with the plasma membrane protein of interest tagged with GFP and fused to four repeats of FM domains, and a furin cleavage site (see Appendix A). The FM domains are variants of FKBP (FK506-binding protein), which are able to reversibly self-aggregate into homodimers that can disaggregate within minutes after addition of the membrane permeable drug DD solubilizer (Takara Bio Inc., Kusatsu City, Japan, Cat Number: 635053). 

Seven DIV culture hippocampal neurons were used for trafficking experiments. Neurons were transfected with TfR-GFP-FM4 or FM4-L1-GFP or FM4-ApoER2-GFP or FM4-p75NTR-GFP plus GalT2-mCherry and BARS mutants or empty vectors. At 12 h post-transfection, the cultures were treated with 100 μg/mL cycloheximide for 1.5 h to prevent newly FM4-GFP-tagged plasma membrane protein synthesis and its entering to the biosynthetic route. Next, neurons were treated with 2.5 µM DD solubilizer (Takara Bio Inc., Cat Number: 635053) and fixed at the indicated times to evaluate the traffic of monitor cargo through the secretory pathway.

### 2.6. Morphometry and Quantitative Fluorescence 

Montages showing the complete neuronal arbor of transfected neurons were created from confocal images (maximal projections) acquired through a 63 × 1.4 NA oil objective lens. Dendritic (MAP2+) and axonal (Tau-1+) neurites shape parameters were measured with the morphometric menu of Image J [13,14].

For transport experiments the relative intensities of FM-cargo-GFP in the dendritic process and in the Golgi area delimitated by GalT2 distribution were quantitated and a ratio was calculated using Image J software. First, an average background fluorescence value was determined from several regions containing unlabeled neurites and then subtracted from the whole image. To quantify the fluorescence in dendrites, several one-pixel-wide lines were drawn extending from the base to the tip of dendritic processes using the MAP2 signals as a guide, then these regions were transferred to the cargo-GFP fluorescence image, and the average fluorescence for each region was calculated. In addition, intensity measurements were performed in the Golgi region: a polygon was drawn using the GalT2 signal as a guide and then this region was transferred to the cargo-GFP fluorescence image, and the average fluorescence was calculated. Then, a ratio was calculated between mean average fluorescence intensity in dendrites and the average fluorescence intensity in the Golgi area. This ratio was plotted against ligand exposition time. The dendrite/Golgi ratios for each construct were calculated from a minimum of 20 cells from at least three different cultures. Statistical significance was assessed using Tukey post hoc test.

### 2.7. In Utero Electroporation, Immunohistochemistry 

In utero electroporation was executed as previously described [15] with minor modifications. Briefly, C57BL/6J mice pregnant at 15 days post-fertilization were anaesthetized with isoflurane (Piramal UK). Needles for injection were pulled from P-97 Flaming/Brownglass capillaries (World Precision Instruments, Sarasota, FL, USA). Injection solutions were prepared with a mixture of plasmids according to the experiment: sh-RNA-BARS-GFP; sc-shRNA-BARS-GFP; GFP and BARS S147A-G172E-GFP, at a concentration of 2 μg/μL; sh-RNA-BARS-HcRed + ApoER2-GFP or sc-shRNA-BARS-HcRed + ApoER2-GFP or RFP + ApoER2-GFP or BARS S147A-G172E-RFP + ApoER2-GFP at a concentration of 2 μg/μL and 1 μg/μL, respectively, in sterile water and Tripan blue and then injected in the ventricle of the embryo. With a dedicated electroporator (BTX electroporator, Harvard Apparatus, Holliston, MA, USA) five pulses of 38 V (50 ms ON, 950 ms OFF) were applied on the head of the embryos using 5 mm electrodes. The embryos were returned into the abdominal cavity and the abdominal cavity was sutured. After this procedure, an analgesic, tramadol (5 mg/kg), was administered. Three days after surgical intervention, mice were anesthetized with Ketamine/Xilancine mixture and perfused transcardially with 4% (*w*/*v*) paraformaldehyde (Sigma-Aldrich, Cat Number: 441244) in PBS. The perfused brains were removed and post-fixed in 4% (*w*/*v*) paraformaldehyde at 4 °C overnight. In order to performer coronal sections, the brains were cryoprotected by overnight immersion in 30% (*w*/*v*) sucrose in PBS and embedded in mounting medium for freezing, Cryoplast (Biopack, Bondi Junction, NSW, Australia, Cat. Number: 2000120400). Then, the brains were cryopreserved using liquid nitrogen and stored at −80 °C for two days. Floating cryosections of 50 μm were obtained and permeabilized with PBS containing 0.5% (*v*/*v*) Triton-X 100 and blocked with 2% (*w*/*v*) BSA and 0.3% (*v*/*v*) Triton X-100 in PBS for 1.5 h followed by incubation with primary antibody diluted in blocking buffer at 4 °C overnight. After washing three times with PBS, the sections were incubated with appropriate secondary antibodies diluted in blocking buffer for 1 h at room temperature. The sections were washed with PBS followed by DAPI staining (15 min at RT). 

### 2.8. Quantification of Neuronal Migration, Morphology, and Cargo Distribution

Confocal microscopy of brain slices was performed with an Olympus FV1200 (Olympus FV1200, Japan) confocal microscope with Tilescan. Images were captured and digitized with the Olympus Fluoview Software using a 1024 × 1024 scan format with 20×, 40× and 60× objectives. The images were analyzed using ImageJ software. 

For quantification of radial migration, the cortex was divided into cortical plate (CP), intermedial zone (IZ), and ventricular/subventricular zones (VZ/SVZ). GFP-expressing cells were quantified in a common boxed region through the somatosensory cortex at the same rostrocaudal level for each brain. For the morphological analysis, we quantified the percentage of neurons that present unipolar/bipolar (with a clear leading process), and multipolar morphology was quantitated in the RMZ (upper intermedial zone—cortical plate).

To evaluate the distribution of ApoER2-GFP in cortical neurons, we quantified the relative intensities of the GFP signal in the leading process and in the soma area delimitated using the soluble RFP or HcRed distribution in neurons of the radial migrating zone. The ratio was calculated using Image J software. In all cases at least 3 independent experiments were carried out; at least 20 brain slices from each experiment were quantified. An average of at least 100 cells was scored for each condition.

To evaluate the centrosome position, in electroporated radial medial zone (RMZ) neurons, 60× images were used. We traced a perpendicular line to the cortical plate crossing the center of the cell nucleus, dividing the cell body into two halves, one oriented toward the cortical plate (CP) and the other to the ventricular zone (VZ). Then, the percentage of centrosomes in each half was measured. 

### 2.9. Immunoblotting 

Changes in the levels of BARS during neuronal development or after RNA interference (RNAi) treatment in a mouse cell line B16 were analyzed by Western blotting as described previously [13]. The polyclonal anti-BARS antibody was used diluted 1/1000 and mAb anti-α-Tubulin (Sigma Aldrich, Cat Number AB3201, RRID: AB_177350) was used diluted 1/2000. Densitometry of Western blots were performed using Scion Image software.

### 2.10. Statistical Analyses

All data are presented as mean ± SEM. Statistical significance between conditions was calculated with Prism 5.0 (GraphPad Software) using tests as indicated in each figure. In all cases: * *p* < 0.05, ** *p* < 0.01, *** *p* < 0.001. Blind tests, sample calculation, and randomization were not conducted in this study. 

### 2.11. Data Availability

The datasets generated and/or analyzed during this study are available from the corresponding author on request.

## 3. Results

### 3.1. Expression and Subcellular Distribution of BARS in Cultured Hippocampal Neurons

To begin analyzing possible functions of BARS during neuronal development we first examined its pattern of expression and subcellular distribution in rat cultured hippocampal pyramidal neurons, a widely used model to study neuronal polarization and development in vitro [16,17]. Western blot analysis of cell extracts obtained from neurons cultured at different time points (1, 8, and 14 DIV, see Methods) stained with a polyclonal antibody that specifically recognizes BARS revealed that E18 hippocampal cells express BARS at the time of plating, and this remains constant at later time points (Figure 1a). Immunofluorescence staining of young (1–3 DIV) and mature (10–14 DIV) rat cultured neurons showed the presence of BARS in the cell nucleus, the cell body, and neurites. Confocal microscopy revealed that BARS immunolabeling localizes to punctate (vesicle-like structures) located close to the plasma membrane (Figure 1b, arrows) and within cell bodies and neurites, presumably both axons and dendrites (Figure 1b,c); labeling was also observed in the perinuclear region where it localized with some but not all Golgi structures, visualized by GM130 (Figure 1c–e, arrows) or TGN-38 (not shown) labeling.

### 3.2. BARS Regulates Axon-Dendrite Extension in Cultured Hippocampal Neurons

In the next series of experiments, we analyzed the consequences of the up- or down-regulation of BARS expression on axon-dendrite growth (e.g., elongation and branching). We first evaluated the effect of BARS suppression; plasmids containing a sh-RNA-BARS-GFP or scrambled sc-shRNA-BARS-GFP (Materials and Methods; see also [18]) were used to transfect rat primary neuronal cultures shortly after plating or after 5 days in vitro (DIV). Cultures were fixed 48 h after transfection, stained for BARS and markers for either axons (e.g., Tau-1) or dendrites (e.g., MAP2), and neuronal morphology was quantitated in maximal projection images obtained by confocal microscopy. Immunoblotting and immunofluorescence revealed that transfection with shRNA-BARS-GFP reduced BARS expression (Appendix A). 

As reported previously [16,18], after 2 DIV most control neurons (either non-transfected or expressing control sc-shRNA-BARS-GFP) have acquired morphological polarity displaying a single long (more than 250 µm in length) Tau-1 (+) axon (Figure 2a,b,g) and several much shorter Tau-1 (−) minor neurites. Neurons expressing the shRNA-BARS-GFP displayed a similar morphology, but with axons significantly shorter than those of control cells with less collateral branches (Figure 2c,d,g). By contrast, ectopic expression of BARS WT significantly increased axonal and minor neurite lengths, with both types of processes exhibiting numerous collateral branches (Figure 2e–g). BARS up- or down-regulation also influenced dendritic morphology. BARS suppression significantly reduced dendritic length and branching while overexpression had the opposite effect (Figure 2h–q,s). To gain further insight into this phenomenon, we tested whether it might reflect BARS functioning at the Golgi apparatus by examining the consequences of overexpressing BARS mutants (Appendix A and Methods) with reduced nuclear localization or lacking fission activity on dendritic morphology. We first tested BARS G172E, a mutant variant exhibiting loss of repressor activity caused by failure to dimerize and localize to the cell nucleus [9]; ectopic expression of this mutant increased dendritic length and branching, as in the case of the WT variant (Figure 2r,s). Then, we evaluated the effect of two BARS mutants with reduced fission activity and both of them, BARS S147A [4] (Figure 2s) and BARS D355A [3] (not shown), significantly decreased dendritic length. The stimulatory effect of BARS was also absent after expression of the double mutant S147A G172E (Figure 2s). Together, our observations indicate that BARS promotes axon/dendrite growth and branching and suggest that this effect involves Golgi membrane fission. 

### 3.3. BARS Regulates Neuronal Migration and Multipolar-to-Bipolar Transition in Cortical Neurons That Develop In Situ

To follow up with these observations, in the next series of experiments we used in utero electroporation (IUE) of embryonic mouse brains (E15.5) with plasmids encoding the shRNA-BARS-GFP (or its scrambled variant), or the fission mutant BARS S147A, or the double mutant BARS S147A G172E (a monomeric cytoplasmic fission dominant-negative and co-repressor activity dominant-negative mutant) to assess the consequences of BARS down-regulation on cortical migration and polarization (Figure 3). Three days after IUE (E18.5), embryos were sacrificed and GFP-expressing brains fixed to visualize the cerebral cortex. Cortical development was evaluated as described by Kriegstein and Noctor [19]. During this time frame, migrating cortical neurons undergo a multipolar–bipolar transformation, which has been considered equivalent to the stage 2–3 switch observed in cultured neurons (see below). The results obtained showed that most shRNA-BARS-GFP-expressing cells failed to migrate and reach upper cortical layers, either the intermediate zone (IZ) or the cortical plate (CP); more than 70% of them remained arrested at the subventricular zone (SVZ) (Figure 3a). Neurons expressing BARS S147A + GFP (not shown) or BARS S147A G172E + GFP also failed to reach the CP, with most of them (around 60%) halted at the IZ (Figure 3b). The migration defect observed in the down-regulation of BARS is independent of neuronal differentiation and proliferation since there are no major changes in the percentage of electroporated cells (GFP^+^) stained with the early neuronal marker βIII-tubulin or the proliferation marker Ki67 antibodies between control scrambled shRNA and shRNA-BARS (Appendix A). We also analyzed the position and orientation of centrosomes using a γ-tubulin antibody in electroporated migrating neurons, a future needed for axon formation in situ [20]. The results obtained showed that the centrosome position orients normally in BARS down-regulated neurons (Appendix A). These results suggest that BARS regulates the migration and multipolar-to-bipolar transition without affecting proliferation, differentiation, and centrosome orientation. 

In control brains, most neurons located at the SVZ and lower IZ (multipolar migration zone, MMZ) displayed a multipolar morphology, resembling stage 2 of polarity observed in culture [21], while those located at the upper IZ and cortical plate (radial migration zone, RMZ) exhibited a bipolar/polarized morphology with oriented processes, one facing the CP (future dendrite) and the opposite (future axon) towards the ventricular zone (VZ) [22,23,24]. Therefore, we estimated the percentage of multipolar and bipolar neurons at the RMZ after down-regulation of BARS functioning. The results obtained show that many (around 50%) neurons located at this zone and expressing either the shRNA-BARS or the BARS double mutant S147A G172E displayed a multipolar morphology as opposed to the very small number detected in control brains (Figure 3c,d). 

Taken together, these results indicate that BARS regulates multipolar-to-bipolar transition and neuronal migration in cortical neurons that develop in situ, and most likely involve BARS activity in Golgi membrane fission. 

### 3.4. BARS Regulates Membrane Trafficking in Developing Neurons in Culture and In Situ

The results described in previous sections raised the possibility that BARS might regulate axon/dendrite growth by controlling membrane fission at the Golgi apparatus (GA). Therefore, we evaluated if BARS regulates the exit of neuronal membrane proteins from this organelle. To this end, we ectopically expressed prototypical dendritic (Transferrin Receptor, TfR, and Apolipoprotein E Receptor 2, ApoER2) and axonal (L1 cell adhesion molecule, L1, and p75 neurotrophin receptor, p75^NTR^) membrane proteins. Recombinant proteins were tagged with FM reversible self-aggregating domains (Appendix A) that result in their accumulation at the endoplasmic reticulum (ER). Addition of the cell permeable DD solubilizer [25,26] (Appendix A) dissociates the FM tag, allowing synchronic monitoring in time and space traffic of dendritic and axonal membrane proteins along the exocytic pathway. Seven DIV neurons were transfected with different FM-containing constructs (Appendix A) and 12 h later cultures treated with cycloheximide for 90 min followed by treatment with DD solubilizer; cultures were then fixed at different time intervals ranging from 0–300 min (Appendix A). 

TfR-GFP-FM4 displayed a diffuse distribution throughout the cell body, and dendrites (Figure 4a) in cultured neurons fixed before and immediately after the addition of cycloheximide or DD; this pattern was similar to that of ER markers, such as calnexin (not shown). However, 15 min after treatment with DD, TfR-GFP-FM4 redistributed to the cell body completely colocalizing with GA markers, such as TGN38 (Figure 4b); by 60 min punctate labeling was also detected in dendritic-like processes (Figure 4c). Four hours later (300 min after DD), TfR-GFP-FM4 almost disappeared from the TGN, with strong labeling of dendritic processes (Figure 4d); GFP fluorescence was also detected in close proximity to the plasma membrane (Figure 4d, arrowheads). TfR-GFP-FM4 distribution was also evaluated in neurons co-expressing BARS D355A (Figure 4e–h). The results obtained showed that ectopic expression of this fission-defective mutant did not prevent accumulation of TfR-GFP-FM4 at the GA 15 min after DD (Figure 4f), but considerably decreased its presence in dendrites at later time points; thus, at 60 and 300 min post-solubilizer, most of the TfR-GFP-FM4 signal remained at the GA (Figure 4g,h). Quantitative evaluation of TfR-GFP-FM4 fluorescence ratio between dendrites and the GA in control conditions and in cells expressing the fission-deficient BARS mutant D355A confirmed our qualitative observations (Figure 4i). As already at 60 min after DD solubilizer was shown a markedly and statistically significant difference between control and BARS D355A fission mutant; we used this time point to evaluate forthcoming treatments (Figure 4i). A similar phenomenon was observed when co-expressing TfR-GFP-FM4 with BARS S147A (Figure 4j–l,s) or with BARS double mutant S147A G172E (Figure 4m–o,s). By contrast, ectopic expression of BARS WT accelerated and significantly increased the GA exit and trafficking of TfR into dendrites. (Figure 4p–s). Taken together, the experiments described in this section clearly demonstrate that BARS regulates the TfR exit from the GA, most likely via the TGN.

We then evaluated trafficking of another dendritic membrane protein, namely the Apolipoprotein E Receptor 2 (ApoER2) that has been implicated in cortical layer development, dendritic growth, spine formation, and synaptic plasticity. In rat cultured hippocampal pyramidal neurons, ectopically-expressed ApoER2-GFP (Appendix A) localized to dendrites; FM4-ApoER2-GFP behaved as the TfR, moving from the ER to the TGN within 15 min after addition of DD (Figure 5a,b) and into dendrites at later time points (Figure 5c). Interestingly, co-expression of BARS S147A halts trafficking of FM4-ApoER2-GFP into dendrites without altering ER to GA passage (Figure 5d–f). To complement these experiments, we used IUE to evaluate the distribution of ApoER2-GFP in cortical neurons that develop in situ and express sc-shRNA-BARS-HcRED or shRNA-BARS-HcRED or the double mutant BARS S147A G172E-RFP. The results obtained show that either down-regulation of BARS expression or of its fission activity results in failure of ApoER2 to enter into neurites; in these cells most of the ApoER2-GFP signal localized to a rather limited region of the cell body resembling the GA, as opposed to control neurons that display fluorescence not only in the cell body but also in the apical oriented neurite or future dendrite (Figure 5h–m and Figure 5h’–k’ for high magnification views).

In the final set of experiments, we explored if trafficking of axonal membrane proteins was also altered in rat cultured neurons expressing fission-defective mutants of BARS; for such a purpose we first analyzed trafficking of L1, a cell adhesion molecule that is transported in carriers that enter axons and dendrites but that only fuse with the axonal membrane [13,27,28,29]. The results obtained showed that FM4-L1-GFP accumulated at the TGN 15 min after addition of DD, entering dendrites at later time points (Appendix A). By contrast, in neurons co-transfected with FM4-L1-GFP + BARS S147A, most of the labeling remained at the TGN, after addition of DD solubilizer (Appendix A). We also evaluated p75^NTR^, a neuronal membrane protein that also traffics to axons and dendrites and that is preferentially, but not exclusively, delivered to the axonal surface [27,30]. No alterations in p75^NTR^ transport from the ER to GA or from the TGN to neurites were detected in neurons expressing BARS fission-defective mutants (Appendix A). Similar results have been observed in COS and polarized MDCK cells, where inhibition of BARS does not affect the apical trafficking of p75^NTR^ [3]. Collectively, these findings provide strong evidence that BARS regulates the exit and trafficking of several neuronal membrane proteins from the GA. Meanwhile, it does not affect the p75^NTR^-containing carrier exit, it markedly delays, to a different extent, the trafficking of dendritic membrane proteins (TfR and ApoER2) and another axonal (L1) membrane protein.

## 4. Discussion

Accurate trafficking of membrane components throughout the secretory pathway is crucial for neuronal migration, axon/dendrite growth, the establishment and maintenance of neuronal polarity, as well as synaptic plasticity [31,32,33,34]. Biogenesis and trafficking of neuronal membrane proteins start at the rough endoplasmic reticulum (RER) and continue along the Golgi apparatus (GA), where axonal and dendritic membrane proteins are sorted and packed into different carriers that will bud and exit at the TGN. They will then be conveyed towards the plasma membrane by different types of molecular motors, including kinesin super-family members and myosins, responsible for long- and short-range transport motions, respectively [35,36]. Despite considerable advances towards understanding major events and mechanisms underlying trafficking of neuronal membrane proteins, the identity of the fission machinery mediating carrier exit from the TGN has remained largely unexplored; we now present evidence indicating that in developing neurons BARS is one such component. 

Earlier studies aimed to examine in mice the consequences of mutations of genes encoding CtBP family members, including BARS, revealed that they are crucially required for proper brain development. It was proposed that the observed phenotypes largely reflect de-regulated gene expression resulting from the absence of CtBP-mediated co-repressor transcriptional activity [1,8]. We now extended these observations, showing that BARS also contributes to nerve cell development by regulating fission of exocytic carriers containing axonal or dendritic membrane proteins involved in migration, growth, and polarization. 

Down-regulation of BARS expression significantly decreased axonal and dendritic growth in cultured hippocampal neurons, as well as migration and multipolar-to-bipolar transition in cortical neurons that develop in situ. The possibility that these phenotypes may result from BARS acting at the GA is supported by the following observations. First, similar phenotypes were detected in neurons expressing BARS mutants with reduced nuclear localization and lacking fission activity. Conversely, expression of BARS WT or BARS G172E increased axonal and dendritic growth, but their effects were prevented by co-expression of fission-defective BARS mutants. Second, expression of these mutants significantly reduced exit from the GA of membrane proteins implicated in neuronal development and plasticity; they comprise two main groups. 

The first one corresponds to proteins sorted into carriers that traffic to dendrites and are largely excluded from axon (e.g., selective sorting and delivery [31]; this study). It includes: (1) TfR, a classic dendritic membrane protein, required for brain iron homeostasis and involved in AMPA receptor recycling and synaptic plasticity [37,38,39]; and (2) ApoER2, a Reelin receptor, directly implicated in neuronal cortical migration, dendritic growth, and the multipolar–bipolar transition [40]. The possibility of BARS mediating fission of TGN-derived carriers containing dendritic membrane proteins is in line with observations in epithelial cells showing that it fissions VSVG-containing basolateral carriers [3,4]; in this regard, it is worth noting that identical motifs mediate transport and delivery of dendritic and basolateral membrane proteins to the cell surface [31]. The second group belongs to carriers containing L1, a cell adhesion molecule that traffics to axons and dendrites but that is selectively delivered to the axonal surface [29,31] and has been implicated in neuronal migration, axon growth, and guidance [41,42]. The exit from the GA of p75^NTR^, a membrane protein that is preferentially incorporated into the axonal membrane [30] but that is also delivered to the dendritic surface [27], was not altered after expression of BARS fission-defective mutants. Likewise, trafficking of axonal and dendritic proteins from the ER to the GA was not significantly altered after expression of BARS fission-defective mutants; thus, after addition of the solubilizer agent DD, all FM constructs accumulate at the GA within the next 15 min, completely disappearing from non-Golgi areas. 

In polarized epithelial cells (e.g., MDCK cells), LIMK1 and PKD1 are two additional elements of the membrane fission machinery acting at the TGN that regulate the exit of membrane proteins aimed to the apical or basolateral membrane, respectively [43,44]. For example, a LIMK1-actin-cortactin-dynamin-dependent fission machinery mediates exit from the GA of p75^NTR^-containing carriers aimed to the apical surface [43]. LIMK1 and PKD are also expressed in neurons, localized to the GA, and engaged in the dynamin-mediated fission of Golgi-derived tubules and the generation of Golgi outposts [10,13,14]. LIMK1 has also been implicated in regulating Golgi-derived export of synaptophysin-, βGC-, and NCAM- (the chicken homologue of L1) containing vesicles [14]; this last observation raises the possibility that at least for L1, two types of fission events (dynamin-dependent and -independent) might regulate its exit from the GA. 

PKD1 also regulates trafficking of dendritic membrane proteins (e.g., TfR, LRP4); however, PKD down-regulation does not result in the failure of dendritic carriers to exit from the TGN, but in mis-sorting of their cargo into vesicles containing axonal membrane proteins (e.g., VAMP2) distinct to those containing L1 and that traffic to both axons and dendrites [13]. These observations are intriguing, since studies in non-neuronal cells suggest that BARS and PKD1 act in concert by forming part of a protein complex required for fission and the generation of post-Golgi basolateral carriers [4]. Additional studies are now required to analyze the mechanism by which BARS fissions dendritic carriers at the TGN. In addition, it will be worthwhile to explore whether BARS is also involved in the fission of Golgi-derived tubules and the generation of Golgi outpost, as we found that GOPs of minor dendrites is not dependent on dynamin activity [10].

## Figures and Tables

**Figure 1 cells-11-01320-f001:**
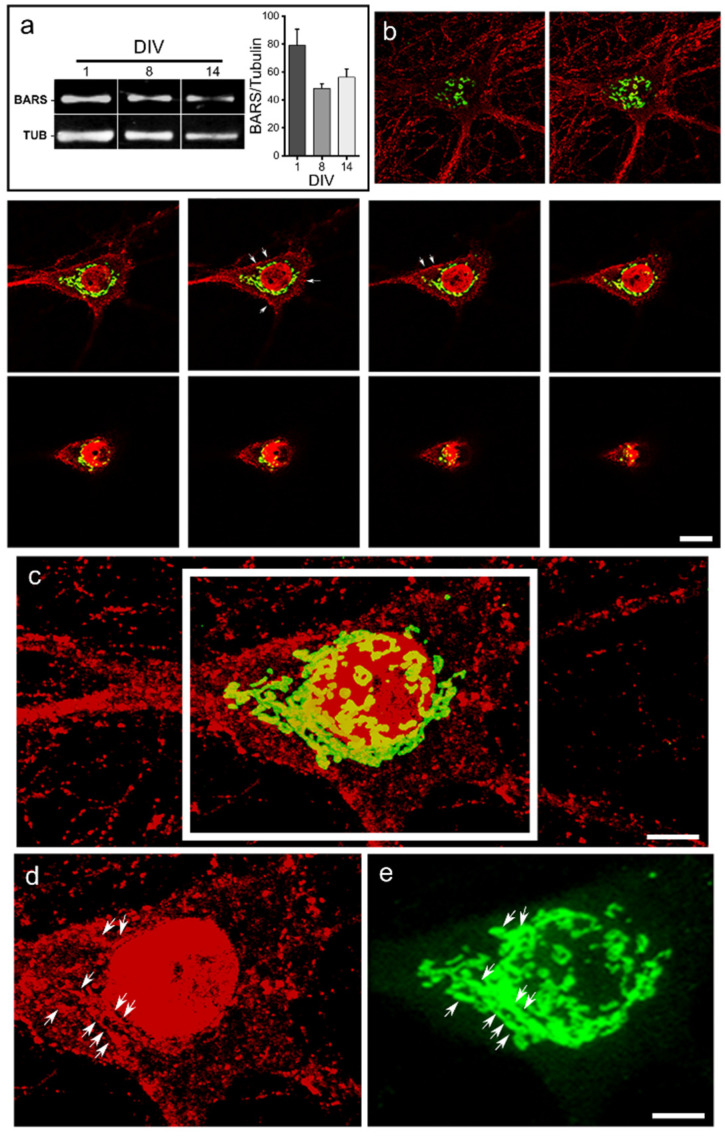
Expression and distribution of BARS in cultured neurons. (**a**) Western blots (left panel) and quantification (right panel) showing the presence of BARS in cell extracts obtained from cultured hippocampal neurons after development for 1, 8, and 14 DIV (days in vitro). TUB: α-tubulin. Graphs represent mean ± S.E.M. of BARS densitometry data normalized to α-tubulin. One-way ANOVA test. Three independent cultures per group were analyzed. (**b**) A *z*-series of confocal images from a 14 DIV cultured hippocampal neuron showing the distribution of BARS (red) and GM130 (green). Note that BARS immunofluorescence localizes to the cell nucleus, the cell soma, and neurites (punctate labeling); labeling is also observed in close proximity to the plasma membrane (arrows) and the Golgi region. Scale bar: 5 µm. (**c**) A maximal projection image collected from the images shown in (**b**). (**d**,**e**) Single channel of the insert in (**c**) showing the distribution of endogenous BARS (**d**, red) and the Golgi marker GM130 (**e**, green). Note that BARS immunolabeling partially colocalizes with GM130 + profiles (arrows). Scale bar: 2.5 µm.

**Figure 2 cells-11-01320-f002:**
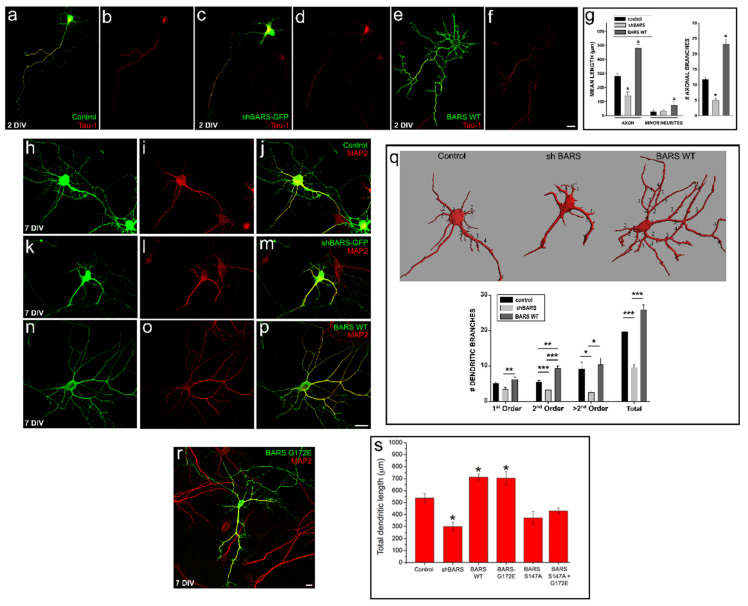
BARS regulates axon and dendrite growth in cultured neurons. (**a**–**f**) Confocal images showing the morphology of cultured hippocampal neurons (2 DIV) transfected with sc-shRNA- BARS-GFP (control), sh-RNA-BARS-GFP (shBARS-GFP), or BARS WT. Cultures were transfected shortly after plating, fixed 48 h later, and stained with the mAb Tau-1 (red) and a polyclonal Ab against BARS protein (green, in (**e**)). Scale bar: 10 µm. (**g**) Graph showing quantification of total axonal and minor neurite lengths (left panel) and the number of axonal branches (right panel) after up- or down-regulation of BARS expression. Graphs represent mean ± S.E.M.; * *p* < 0.05; one-way ANOVA and Tukey’s post hoc test. (**h**–**p**) Confocal images showing dendritic morphology of cultured hippocampal neurons (7 DIV) transfected with sc-shRNA-BARS-GFP (control), sh-RNA-BARS-GFP (shBARS-GFP) or BARS WT. Cultures were transfected 5 days after plating, fixed 48 h later, and stained with an mAb against MAP2 (red) and a polyclonal Ab against BARS protein (green, in (**n**–**p**)). Scale bar: 10 µm. (**q**) Three-dimensional reconstruction images (upper panel) and quantification of dendritic branches (lower panel) of the transfected neurons shown in (**h**,**k**,**n**). Graphs represent mean ± S.E.M.; * *p* < 0.05, ** *p* < 0.01, *** *p* < 0.001; one-way ANOVA and Tukey’s post hoc test. (**r**) Confocal image of a cultured neuron transfected with the BARS mutant G172E + GFP and stained with the mAb Tau-1. Scale bar: 10 µm. (**s**) Graph showing quantification of total dendritic length after up- or down-regulation of BARS levels or expression of BARS mutants. Graphs represent mean ± S.E.M.; * *p* < 0.05; one-way ANOVA and Tukey’s post hoc test. For all experiments, 9 to 15 neurons were quantified pooled from at least three independent cultures.

**Figure 3 cells-11-01320-f003:**
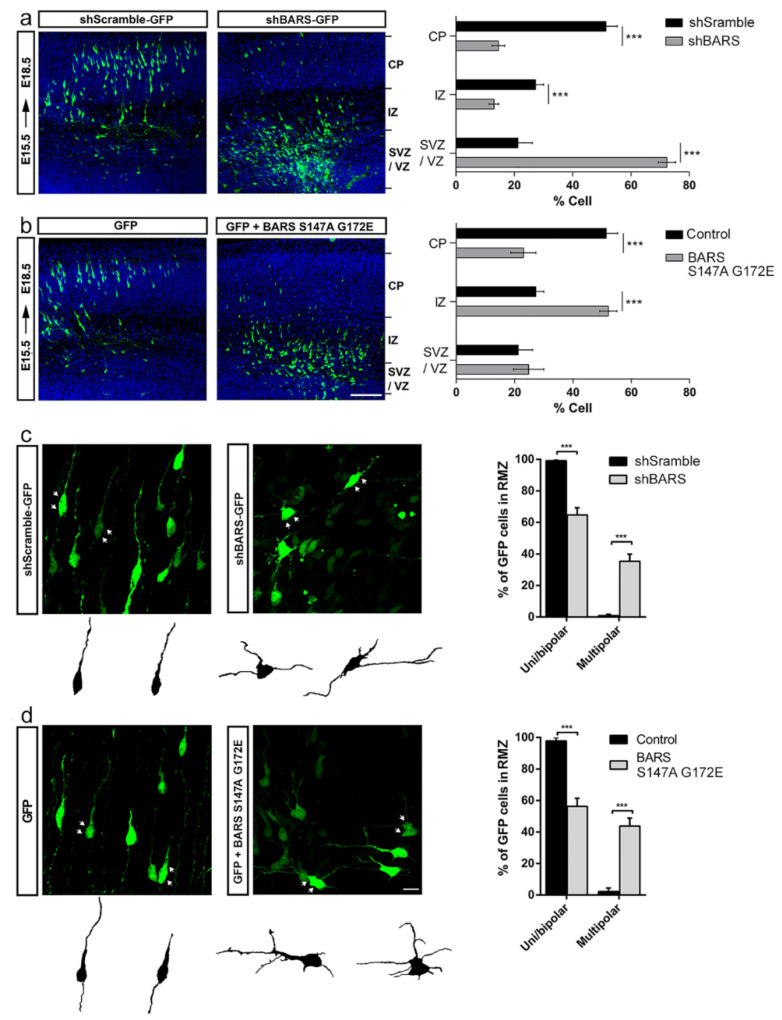
BARS regulates migration and multipolar-to-bipolar transition in cortical neurons in situ. (**a**) Left and middle panels: representative images of coronal cortical slices of mouse brain (embryonic day E18.5), expressing sc-shRNA-BARS-GFP (shScramble) or shRNA-BARS-GFP (shBARS) after IUE; right panel: quantification (%) of GFP-positive electroporated neurons by layers. VZ: ventricular zone, SVZ: subventricular zone, IZ: intermediate zone, CP: cortical plate. (**b**) Idem to previous panels, but after IUE of GFP or GFP + BARS S147A G172E; right panel: quantification (%) of GFP-positive electroporated neurons by layers. VZ: ventricular zone, SVZ: subventricular zone, IZ: intermediate zone, CP: cortical plate. Scale bar 100 µm. (**c**) Left and middle panels: High magnification confocal images of cortical neurons expressing either sc-shRNA-BARS-GFP (shScramble) or shRNA-BARS-GFP (shBARS) at radial migration zone, RMZ; right panel: quantification (%) of uni/bipolar and multipolar morphologies in shScramble- and shBARS-expressing neurons. (**d**) Left and middle panels: high magnification confocal images of cortical neurons expressing either GFP or GFP + BARS S147A G172E. Scale bar: 10 µm; right panel: quantification (%) of uni/bipolar and multipolar in GFP- and BARS S147A G172E-expressing neurons. For all experiments 10 slices were analyzed from 3 independent IUE for each experimental condition. Graphs represent mean ±S.E.M.; *** *p* < 0.001, Student’s *t*-test.

**Figure 4 cells-11-01320-f004:**
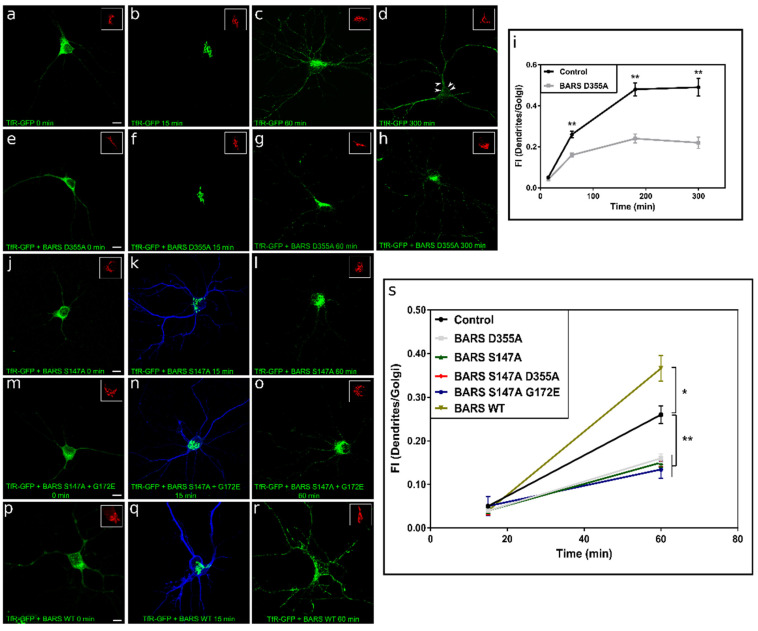
BARS regulates trafficking of TfR. (**a**–**h**) A series of confocal images showing the distribution of ectopically-expressed TfR-GFP-FM4 alone or after co-transfection with BARS D355A. The distribution of TfR-GFP-FM4 is indicated at different time points after the addition of DD solubilizer. Inset, distribution of the Golgi apparatus marker TGN38 (red). In control cells (GFP-transfected) most of the TfR-GFP labeling disappeared from the Golgi region 300 min after the addition of DD (**d**); the arrowheads in (**d**) indicate the presence of the labeling close to the dendritic plasma membrane. By contrast, in neurons co-expressing the BARS fission-defective mutant most of the labeling remains in the Golgi area (**h**). Scale bar: 10 µm. (**i**) Graphs showing the fluorescent intensity (FI) ratio between TfR-GFP labeling in dendrites and the Golgi area at 15, 60, 180, and 300 min after the addition of DD solubilizer in control and in neurons expressing BARS D355A mutant. (**j**–**o**) A series of confocal images showing the distribution of ectopically-expressed TfR-GFP-FM4 after co-transfection with BARS S147A (blue in k) or BARS S147A G172E (blue in n). The distribution of TfR-GFP-FM4 is indicated at different time points after the addition of DD solubilizer. Inset, distribution of the Golgi apparatus marker TGN38 (red). Scale bar: 10 µm. (**p**–**r**) A series of confocal images showing the distribution of ectopically-expressed TfR-GFP-FM4 after co-transfection with BARS WT (blue in (**q**)). The distribution of TfR-GFP-FM4 is indicated at different time points after the addition of DD solubilizer. Inset, distribution of the Golgi apparatus marker TGN38 (red). Scale bar: 10 µm. (**s**) Graphs showing the fluorescent intensity (FI) ratio between TfR-GFP labeling in dendrites and the Golgi area at 15 and 60 min after the addition of DD solubilizer in control and in neurons co-expressing BARS WT or different mutants. For all experiments 9 to 15 neurons were quantified pooled from at least three independent cultures. Graphs represent mean ± S.E.M.; * *p* < 0.05, ** *p* < 0.01; one-way ANOVA and Tukey’s post hoc test.

**Figure 5 cells-11-01320-f005:**
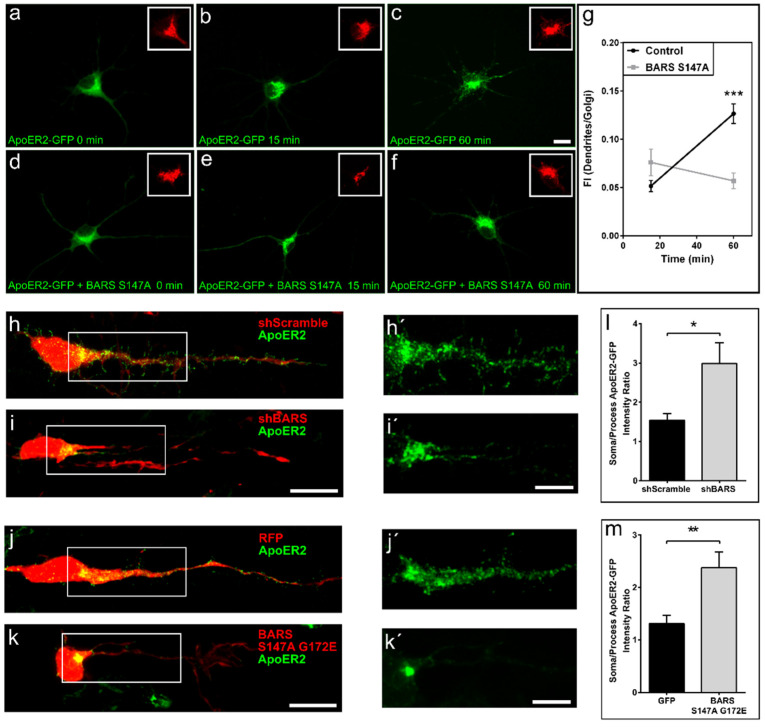
BARS regulates trafficking of ApoER2. (**a**–**f**) A series of confocal images showing the distribution of ectopically-expressed FM4-ApoER2-GFP alone or after co-transfection with BARS S174A and visualized 0, 15 and 60 min after the addition of DD solubilizer. Inset, distribution of the Golgi apparatus marker TGN38. Scale bar: 10 µm. (**g**) Graph showing the fluorescent intensity (FI) ratio between ApoER2-GFP labeling in dendrites and the Golgi area 15 and 60 min after the addition of DD solubilizer in control- and BARS S147A-expressing neurons. For all experiments, 9 neurons were quantified pooled from at least three independent cultures. Graphs represent mean ± S.E.M.; *** *p* < 0.001, Student’s *t*-test. (**h**) A confocal image showing the distribution of ApoER2 in a cortical neuron of mouse brain (embryonic day E18.5) transfected by IUE with sc-shRNA BARS-HcRED (scScramble,) plus ApoER2-GFP; note that the green labeling localized to a discrete area in the cell body and within the apical oriented neurite (image rotated 90º clockwise with respect to the ventral zone–cortical plate axis). (**i**) Idem as in (**h**), but from a mouse brain (embryonic day E18.5) electroporated with sh-RNA-BARS-HcRED (shBARS). (**h’**,**i’**) High magnification views of the inserts shown in (**h**,**i**). Scale bar: 5 µm. (**j**) A confocal image showing the distribution of ApoER2-GFP in a cell from a brain co-electroporated with RFP. (**k**) Idem as in (**j**), but from a brain electroporated with ApoER2 plus BARS S147A G172E. Scale bar: 10 µm. (**j’**,**k’**) High magnification views of the inserts shown in (**j**,**k**). Scale bar: 5 µm. (**l**) Graph showing the fluorescent intensity (FI) ratio between ApoER2-GFP labeling in the soma and the apical oriented neurite in cortical neurons after IUE of shScramble or shBARS. (**m**) Graph showing the fluorescent intensity (FI) ratio between ApoER2-GFP labeling in the soma and the apical oriented neurite in cortical neurons after IUE of RFP or RFP plus BARS S147A G172E. For all experiments 13 to 20 neurons were analyzed from 3 independent IUE for each experimental condition. Graphs represent mean ± S.E.M.; * *p* < 0.05, ** *p* < 0.01, Student’s *t*-test.

## Data Availability

The datasets generated and/or analyzed during this study are available from the corresponding author on request.

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
