# Peer review of "BARS Influences Neuronal Development by Regulation of Post-Golgi Trafficking"

_cells, 2022, doi:10.3390/cells11081320_

Round 1

Reviewer 1 Report

In this study, the authors test BARS protein function in neuronal development due to its role in regulating post-Golgi traffic. BARS is a short form of the CtBP1 protein subfamily, which expresses in specific regions in the adult brain. The deletion of these proteins in mice, together whit the CtBP2 proteins, results in delayed development of the forebrain and midbrain. The author shows the involvement of BARS during neuronal development, reducing the expression of the protein by RNAi. They clearly show that the suppression of BARS inhibit axonal and dendritic elongation in cultured neuron cells and disturb neuronal migration during cortical development in embryos.

The manuscript is well written and laid out. I have no suggestions for changes to the manuscript apart from:

Figure 4. The arrows in picture d are hardly visible, and the blue colour signal in pictures l and k are not explained.

Figure5. it is not clear what the in bracket explanation “(image rotated 90º clockwise)” refers to

Reviewer 2 Report

The manuscript describes that Brefeldin A ADP-Ribosylated Substrate (BARS) is involved in neuronal polarization and migration possible through the membrane trafficking from trans-Golgi network (TGN). The authors showed that knockdown of BARS resulted in shorter axon than control and defects in neuronal migration and morphological changes. Although BARS is a multi-functional protein, they also showed that its dominant negative mutant for membrane trafficking at the TGN disturbed neuronal migration in vivo.

Overall, I felt that this manuscript is informative to many developmental neuroscientists, but I also found several weak points as described below. If the authors adequately address the following issues, I would recommend this manuscript for publication in Cells.

  • The authors claimed that BARS is required for neuronal polarization. However, knockdown of BARS may not affect the formation of neuronal polarity, because the knockdown neurons seemed to possess a single axon. If the neuronal polarity would be disturbed, the knockdown cells should exhibit multiple or no axon(s). The authors should measure the cells with multiple or no axon(s) in Figure 2.

  • In line with my previous comment, the position of the centrosome (or Golgi) should be analyzed in the shBARS-electroporated cells (Figure 3). In addition, the axon length of the electroporated cells should be measured at P4, where control neurons extend their axons to the corpus callosum. If the shBARS-electroporated cells would extend the axons, the authors should change the conclusion from “the neuronal polarization defect” into “defects in the axon elongation and the multipolar-to-bipolar transition of the migrating neurons” in the manuscript.

  • In Figure 1, the colocalization between BARS and TGN38 was not shown. However, this result is important in this manuscript, as the authors showed that BARS is involved in the trafficking from the TGN in Figures 4 and 5. Furthermore, the single channel images of green (GM130/TGN38) and red (BARS) are also required.

  • By using the in vivo electroporation, the plasmids are firstly introduced into the neural progenitors in the ventricular zone. Therefore, the authors should examine whether knockdown of BARS affects the proliferation and neuronal differentiation of the neural progenitors, in addition to the neuronal migration and morphological changes. For example, staining for phosphor-histone H3 (a marker for proliferating cells) and Tuj1 (a marker for neurons) may provide the answer for this question.

Reviewer 3 Report

Brief summary: In this paper, the authors examined the role of BARS (CtBP1-BARS) protein in neuronal development and neuronal trafficking. The manuscript is divided into two parts in which the authors used a toolbox of different BARS mutant/double mutant in rodent neurons. In the first part, the authors examined the effect of BARS mutants and overexpression or down expression of BARS in the dendritic development of rodent neurons in vitro and the polarity of the neuronal population in situ. They observed that the decrease of BARS expression in rodent neurons induces a decrease of dendritic length and an increase of multipolar neurons. In a second part, the authors synchronised the trafficking of 4 different transmembrane proteins (two dendritic and two axonal) in rodent neurons and looked at the effect of BARS mutants in their anterograde trafficking. They observed that the expression of fission dominant-negative BARS delays the anterograde trafficking of TfR, APOE2 and L2 but not p75.

This paper illustrates the involvement of BARS proteins in neuronal development and neuronal anterograde intracellular trafficking in relevant cells (primary rodent neurons). Nonetheless, some controls are missing to support the conclusion fully.

Comments:

Introduction

Line 64-65: “… we have examined the consequences of the up-or down-regulation of BARS on neuronal development and the trafficking of axonal and dendritic membrane proteins.” The authors did not show or discuss the up-regulation of BARS on trafficking while it is needed in the story.

From the Materials and Methods, it looks like the authors used both rat and mouse primary neurons. Nonetheless, it is not always clear which model is used in which experiment. It would add clarity if the authors specify in the Figure legends and/or in the text which model they are using.

Results:

Part 1: Neuronal development

In Figure 1, the authors present BARS expression level at DIV 1 and DIV8 (by western blot) while the immunofluorescence is at DIV14. Also, a quantification would be nice to see if the BARS expression level changes over time. On b, it is hard to see the BARS/Golgi apparatus colocalisation. The authors may consider presenting the two colours separately for the maximal projection (Figure 1C). The authors need to homogenise the size of the pictures.

Figure S1: Western blot’s label in a is blurry. MAP2 is not shown but still in the figure legend (b-d). The scale bar (30 um) doesn’t seem right in d(?)

Line 304:”… to transfect primary neuronal cultures shortly after plating or after 5 days in vitro (DIV)”. It seems that only results from a “transfection shortly after plating” are shown in Figure 2. If this is true, the authors need to amend the text accordingly. Nonetheless, it will be interesting to discuss if there is a difference when neurons are transfected at a later stage, as stated (DIV5).

In Figure 2, another parameter that the authors should consider quantifying is the “number of neurites” and/or the neuronal arborisation as with the measure of the total dendritic length, it is not clear if the high number is due to longer extensions or an accumulation of smaller dendrites.

In part 3.3, the authors observed the role of BARS in the regulation of neuronal migration and polarization in cortical neurons in situ. Additional data (in Figure 3) would help better understand BARS' role. For example, it is unclear why the authors choose to test the double mutation fission defective and dimerization defective. It would be interesting to see the effect of the single fission defective mutant and/or the double fission defective mutant. Also, as a control, it is important to test how the over-expression of BARS WT influences the dynamic of neurons in situ and their morphology/polarity.

The authors might want to add a short comment/conclusion at the end of part 3.3.

Moreover, down-regulation of BARS (shBARS) induces a multi-polarization of the neurons (for 35% of them), and the picture shows elongated ramifications. These results in situ don’t seem consistent with Figure 2, where downregulation of BARS induces a reduction of dendritic length. Also, could the author comment on the fact that the effect on the dynamic of the neurons is different after shBARS or the expression of BARS S147A G172E (shBARS induces a drastic change in the localisation of the neurons whereas BARS S147A G172E induces a moderate change), but the number of multipolar neurons seems to be the same in both conditions.

Part 2: anterograde trafficking

This part is interesting but could be organised better. Controls and information are missing while they are needed to strengthen the conclusion.

I would suggest that the authors follow the anterograde trafficking of some of these cargoes in live cells (together with a transfected Golgi marker). Live-cell imaging will help to directly link the over or down expression of BARS and its effect on trafficking. It will help to define if the intracellular trafficking block is intra-golgien or specific to the TGN exit. Also, it would be interesting to see if they can see some elongation from the TGN when the fission of BARS is downregulated.

Line 394-397: Authors should directly introduce the 4 different transmembrane proteins they will study in this paragraph (TfR, L1 and APOE2 and p75). The authors could add APOE2 and p75 to the schema in Fig Supp 3. Clarity can be added to the schema by using a common legend for all the constructs and adding each protein's orientation.

It is not immediately clear how the authors synchronised the anterograde trafficking of their proteins of interest. It would be useful to add one or two sentences to describe better the synchronisation method used in the paper and why they are using 90 min of cycloheximide before the induction of trafficking (in the introduction or the method).

To fully follow the entry and exit of TfR in the Golgi apparatus, it would be helpful to quantify the colocalisation coefficient between the cargo (TfR) and the Golgi apparatus marker TGN38 (Figure 4). A cis-Golgi marker can be used, and a comparison of the colocalisation coefficients would help to determine if we are in the presence of an intra-Golgi block or a TGN exit block.

One important control missing is the effect of the overexpression of BARS WT on TfR trafficking. Do we observe an acceleration of the trafficking?

The authors should add the time 0 (ER retention) and 300 min for S147A and S147A + G172E. They are missing but are important specifically in the quantification (Figure 4 n).

Cell in Figure 4h might not be representative as the expression of D355A should decrease the neurite length (state line 328 but data not shown).

Minor comment for Figure 4: the zoom for the Golgi apparatus (in red) is small.

It is unclear what is in blue in I and k (please, complete the legend).

It is hard to differentiate the different labels in graph n. The authors might want to use different colours to add clarity.

APOE2 and Figure 5:

The authors stated lines 446-447 that “… trafficking of FM4-ApoER2-GFP into dendrites without altering ER to GA passage (Figure 5c-e)” but the time point 0 min (retention in the ER) is missing (both in the picture and in the quantification). This time point is important and needs to be added as it is the beginning of the anterograde trafficking and the starting point of the experiment. A quantification with a Golgi marker would help to describe the APOE2 Golgi block (see comments for TfR).

Please, homogenise the nomenclature for the protein (either APOE2 or APOEII).

L2 and Figure S5:

Legend and figure labels need to be completed to explain what is blue (and red).

Time point 0 minutes and 300 minutes are missing. The 300 min time point is specifically important here to see if it is a delay or a real block of L2 trafficking.

The Golgi apparatus showed in h seems to be affected by BARS S147A expression; is that a common feature?

The GalT2-GFP + L1-GFP label doesn’t seem right (GalT2-cherry?). Please, correct it in the legend and figure label. The BARS S147A quantification is difficult to see as it is very clear.

p75NTRand Figure S6:

Same comments for p75 (Figure S6): time point 0 and 300 min are missing and need to be added to support the author's conclusion: “No alteration in p75NTRtransport from ER to GA or from the TGN to neurites were detected….” (Line 493-494). Authors could add that this was previously observed in non-neuronal cells (ref 3, Bonazzi et al.).

The GalT2-GFP + L1-GFP label doesn’t seem right (GalT2-cherry?). Please, correct it in the legend and figure label.

The authors could homogenise the presentation of the figures (some are black and white (i.e., Fig 5 and Supp Fig. 6), others are in colours (Fig. 4 and Supp Fig. 5).

The authors might want to add a short comment/conclusion to summarize their interesting findings in the role of BARS in the anterograde trafficking of 4 different proteins. TfR seems to be delayed, whereas APOE2 seems to be blocked. Can the authors comment on that?

Discussion:

The authors might want to comment on the role of BARS in Golgi outposts?

Round 2

Reviewer 2 Report

I have no further comments.